# Bootstrap Aggregation for Model Selection in the Model-free Formalism

Timothy Crawley[1] and Arthur G. Palmer, III[1]

[1]Department of Biochemistry and Molecular Biophysics, Columbia University, 630 West 168th Street, New York, NY 10032, United States

**Correspondence:** Arthur G. Palmer, III (agp6@columbia.edu)

**Abstract.** The ability to make robust inferences about the dynamics of biological macromolecules using NMR spectroscopy depends heavily on the application of appropriate theoretical models for nuclear spin relaxation. Data analysis for NMR laboratory-frame relaxation experiments typically involves selecting one of several model-free spectral density functions using a bias-corrected fitness test. Here, advances in statistical model selection theory, termed bootstrap aggregation or bagging, are applied to $^{15}$N spin relaxation data, developing a multimodel inference solution to the model-free selection problem. The approach is illustrated using data sets recorded at four static magnetic fields for the bZip domain of the *S. cerevisiae* transcription factor GCN4.

## 1 Introduction

Since the original publications in the early 1980's, the model-free formalism of Lipari and Szabo (Lipari and Szabo, 1982a, b) and the related two-step approach of Halle and Wennenström (Halle and Wennerström, 1981) have served as starting points for extracting dynamical information about macromolecules from NMR spin relaxation data. The original approaches represented intramolecular dynamics using a single generalized order parameter and effective correlation time. In the ensuing decades, increasingly complex models have offered a more refined understanding of internal and overall molecular motions. Extended model-free formalisms characterize intramolecular dynamics using generalized order parameters and effective correlation times for more than one (usually two) time scales (Clore et al., 1990; Gill et al., 2016). Related approaches employ discrete or continuous distributions to more fully capture the range of intramolecular correlation times (Lemaster, 1995; Calandrini et al., 2010; Khan et al., 2015; Hsu et al., 2018, 2020; Smith et al., 2019). Other strategies employ physical models or atomistic molecular dynamics simulations for overall rotational diffusion and internal conformational fluctuations, to more directly link the NMR phenomena to underlying physical processes (Tugarinov et al., 2001; Zerbetto et al., 2013; Ollila et al., 2018; Polimeno et al., 2019a, b; Mendelman et al., 2020; Mendelman and Meirovitch, 2021). The availability of extended model-free formalisms, or other approaches with variable numbers of parameters, has created a further dilemma: should a data analysis protocol extract the most exacting information justified by the data, or employ the model most robust to experimental variation.

Several authors have addressed model selection by employing the principle of parsimony or Occam's Razor (Palmer et al., 1991; Stone et al., 1992; Mandel et al., 1995; d'Auvergne and Gooley, 2003; Chen et al., 2004). These approaches seek

to identify the simplest model that explains the data within experimental uncertainties by applying various bias-correcting penalties to the fitness statistic, e.g. F-statistic, Akaike Information Criterion (AIC) or Bayesian Information Criterion (BIC). These corrections alone often fall short of producing robust inferences and may yield parameter values susceptible to instability in both simulated and real-world replicates. In these situations, the model-selection process has failed the principle of 'worrying selectively'. This criterion suggests, "Since all models are wrong the scientist must be alert to what is importantly wrong." (Box, 1976).

To illustrate the issue more concretely, a typical data analysis protocol uses a non-linear weighted least-squares algorithm to fit experimental spin relaxation data with a set of model-free spectral density functions (Mandel et al., 1995; Gill et al., 2016). The resulting $\chi^2$ residual sum-of-squares variables are penalized for the number of adjustable parameters in each model function, the model with the lowest penalized residual sum-of-squares is selected as optimal, and the best-fit parameters of the model reported. However, this procedure is subject to model-selection error: random statistical variation in the experimental data may lead to one model chosen as optimal for a given data set, but another model, with different set of parameters, may be selected if the experimental data were replicated, with consequent different random variation. The problem of joint model-selection and parameter estimation has been explored elegantly by d'Auvergne and Gooley (d'Auvergne and Gooley, 2007, 2008a, b) and by Abergel and coworkers (Abergel et al., 2014).

The present paper addresses model-selection error by using the approach of bootstrap aggregation or bagging. This concept originated from a desire to improve the performance of machine learning algorithms. Thus, Breiman showed that predictor accuracy and stability improved when averaging predictor values obtained from bootstrap replicates of the original training set (Breiman, 1996). Buja and Stuetzle subsequently extended the use of bagging to generalized statistical analysis and showed sampling with and without replacement yield equivalent improvements (Buja and Stuetzle, 2006). The approach and notation of Efron is used in the following (Efron, 2014).

Bootstrap aggregation improves parameter stability; consequently, the resulting variations in model-free parameter values, for example between atomic sites or functional states in a given macromolecule, are more likely to be biologically or chemically meaningful. Although applicable to most model-selection situations, bootstrap aggregation exhibits the most pronounced benefits when the data justify two distinct models with similar degrees of certainty.

Bootstrap aggregation for model-free analysis of NMR spin relaxation relaxation rate constants is illustrated by application to backbone amide $^{15}$N spin relaxation data that have been recorded at $^1$H magnetic fields of 600, 700, 800, and 900 MHz for the bZip domain of the *S. cerevisiae* transcription factor GCN4 by Gill and coworkers (Gill et al., 2016).

## 2 Theory

In the following, the notation used by Efron is rephrased in terms appropriate for NMR spin relaxation data (Efron, 2014). Laboratory-frame nuclear spin relaxation rate constants for backbone $^{15}$N spins can be transformed into sets of spectral density function values, $J(\omega)$, in which $\omega$ is an eigenfrequency of the spin system (Farrow et al., 1995; Gill et al., 2016). Laboratory-frame $^{15}$N relaxation rate constants, typically $R_1$, $R_2$, and the steady-state nuclear Overhauser enhancement (NOE), recorded

at a single static magnetic field yield estimates of $J(0)$, $J(\omega_N)$, and $J(0.87\omega_H)$, in which $\omega_N$ and $\omega_H$ are the ${}^{15}$N and ${}^{1}$H Larmor frequencies. Thus, the number of spectral density values $N = 3G$ in which $G$ is the number of static magnetic fields utilized. In the present application, $G = 4$. The set of experimental spectral densities is described using the following notation:

$$\mathbf{y} = \{y_j\} = \{y_1, y_2, ..., y_N\}, \tag{1}$$

in which the $y_j = J(\omega_j)$ are ordered in increasing values of $\omega$. The values of $J(0)$ are ordered additionally by increasing values of the static magnetic field. The experimental data sets utilized in the present work are not affected by chemical exchange contributions to spin relaxation, but such contributions can be taken into account from the field dependence of transverse relaxation rate constants prior to the model-free analysis (Kroenke et al., 1998).

The extended model-free spectral density function used to fit ${}^{15}$N spin relaxation data is given by the following:

$$J(\omega) = \frac{2}{5} \left[ \frac{S_f^2 S_s^2 \tau_m}{(1 + \omega^2 \tau_m^2)} + \frac{S_f^2 (1 - S_s^2)\tau_1}{(1 + \omega^2 \tau_1^2)} \right.$$
$$\left. + \frac{(1 - S_f^2) S_s^2 \tau_2}{(1 + \omega^2 \tau_2^2)} + \frac{(1 - S_f^2)(1 - S_s^2)\tau_3}{(1 + \omega^2 \tau_3^2)} \right] \tag{2}$$

in which $\tau_1^{-1} = \tau_m^{-1} + \tau_s^{-1}$, $\tau_2^{-1} = \tau_m^{-1} + \tau_f^{-1}$, and $\tau_3^{-1} = \tau_m^{-1} + \tau_s^{-1} + \tau_f^{-1}$ and $\tau_f < \tau_s$. The set of possible model parameters in this function are given by the following:

$$\mu = \{\mu_k\} = \{\tau_m, S_f^2, S_s^2, \tau_f, \tau_s\}, \tag{3}$$

in which $\tau_m$ is the (effective) overall rotational correlation time, $S_f^2$ is the square of the generalized order parameter for internal motions on a fast ($\tau_f \leq 150$ ps) time scale, and $S_s^2$ is the square of the generalized order parameter for internal motions on a slow ($\tau_s > 150$ ps) time scale (vide infra). The square of the generalized order parameter $S^2 = S_f^2 S_s^2$. Overall rotational diffusion has been assumed to be isotropic for simplicity; this assumption can be relaxed as needed (Lee et al., 1997). The spectral density data are fit with a set of nested models. The full model, 'model 5', contains all five parameters, while simpler models, models 1-4, are generated by fixing the value of one or more parameters, effectively removing such parameters from the model. Thus:

Model 1: $\mu = \{\tau_m, S_f^2, 1, 0, 0\}$
Model 2: $\mu = \{\tau_m, S_f^2, 1, \tau_f, 0\}$
Model 3: $\mu = \{\tau_m, 1, S_s^2, 0, \tau_s\}$
Model 4: $\mu = \{\tau_m, S_f^2, S_s^2, 0, \tau_s\}$
Model 5: $\mu = \{\tau_m, S_f^2, S_s^2, \tau_f, \tau_s\}$.

The optimal model $t_1$ and associated parameter values $\mu$ are obtained as follows:

$$\hat{\mu} = \{\hat{\mu}_k\} = t_1(\mathbf{y}), \tag{4}$$

using the lowest penalized residual sum-of-squares as described above. In the present work, the small-sample $AIC_C$ criterion was used for model selection (Hurvich and Tsai, 1989).

In general, a non-parametric bootstrap sample is generated by draws with replacement from the original data $\mathbf{y}$ and defined as follows:

$$\mathbf{y}_i^* = \{y_{ij}^*\} = \{y_{i1}^*, y_{i2}^*, ..., y_{iN}^*\}, \tag{5}$$

in which $i = 1, ... B$ and $B$ is the total number of bootstrap samples. The nature of spectral density data requires care in generating bootstrap samples and the particular procedure employed in the present work is described in Methods.

A conventional non-parametric bootstrap determination of the standard deviations of the parameters $\hat{\mu}$ begins by determining fitted parameters for the $ith$ bootstrap sample as follows:

$$\hat{\mu}_i^* = \{\hat{\mu}_{ik}^*\} = t_1(\mathbf{y}_i^*), \tag{6}$$

in which the fitting model is fixed to the optimal model selected in fitting the original spectral density values and only model parameter values are optimized. The bootstrap estimate of the standard deviation for the $kth$ parameter is derived from the following expressions:

$$\hat{\mu}_k^* = \frac{1}{B} \sum_{i=1}^{B} \hat{\mu}_{ik}^*, \tag{7}$$

$$\hat{\sigma}_k^* = \left[ \frac{1}{B-1} \sum_{i=1}^{B} (\hat{\mu}_{ik}^* - \hat{\mu}_k^*)^2 \right]^{1/2}. \tag{8}$$

In the conventional approach, the reported results of the data analysis are $\{\hat{\mu}_k\}$ and $\{\hat{\sigma}_k^*\}$. Model-selection error is not assessed. This form of bootstrap simulation is an alternative to Monte Carlo simulations to determine parameter uncertanties, which could be regarded as parametric bootstrap simulations (vide infra).

In contrast to the conventional procedure, bootstrap aggregation determines both the optimal fitted model and associated model parameters for each bootstrap sample. Thus, the optimal model $t_i$ is determined for the $ith$ bootstrap sample using the same model selection strategy as for the original data as follows:

$$\tilde{\mu}_i^* = \{\tilde{\mu}_{ik}^*\} = t_i(\mathbf{y}_i^*). \tag{9}$$

Unlike the conventional bootstrap procedure, the different members of the set $\tilde{\mu}_i^*$ obtained by bootstrap aggregation represent different models as well as different sets of optimized parameters. The aggregated, or smoothed, estimator of the $kth$ model parameter is given by the following:

$$\tilde{\mu}_k = \frac{1}{B} \sum_{i=1}^{B} \tilde{\mu}_{ik}^*. \tag{10}$$

To make the above formalism concrete, suppose that for a given set of spectral density values, model selection and parameter optimization for $B$ bootstrap samples yields $B_2$ samples in which model 2 is optimal and $B_3$ samples in which model 3 is optimal, with $B = B_2 + B_3$. The bootstrap aggregated estimates of $\tilde{S}_f^2$ and $\tilde{\tau}_f$ are given by the following:

$$\tilde{S}_f^2 = \frac{1}{B} \left[ \sum_{i \in B_2} \tilde{S}_{fi}^{2*} + \sum_{i \in B_3} 1 \right], \tag{11}$$

$$\tilde{\tau}_f = \frac{1}{B} \left[ \sum_{i \in B_2} \tilde{\tau}_{fi}^{*} + \sum_{i \in B_3} 0 \right]. \tag{12}$$

because model 3 fixes $S_f^2 = 1$ and $\tau_f = 0$. As another example, suppose that for a given set of spectral density values, model selection and parameter optimization for $B$ bootstrap samples yields $B_4$ samples in which model 4 is optimal and $B_5$ samples in which model 5 is optimal, with $B = B_4 + B_5$. The the bootstrap aggregated estimates of $\tilde{S}_f^2$ and $\tilde{\tau}_f$ are given by the following:

$$\tilde{S}_f^2 = \frac{1}{B} \sum_{i=1}^{B} \tilde{S}_{fi}^{2*}, \tag{13}$$

$$\tilde{\tau}_f^2 = \frac{1}{B} \left[ \sum_{i \in B_4} 0 + \sum_{i \in B_5} \tilde{\tau}_{fi}^{2*} \right]. \tag{14}$$

because both models 4 and 5 fit $S_f^2$ as a parameter, but model 4 fixes $\tau_f = 0$.

A smoothed standard deviation for $\tilde{\mu}$ can be obtained using the plug-in-principle (Efron, 2014). Here, the cumulative distribution function for the parameters of interest are estimated using the empirical distribution function of the bootstrap replicates. Using the above notation, the number of times that the $ith$ bootstrap replicate, $\mathbf{y}_i^*$, contains the spectral density value $y_j$ is given by the following:

$$Y_{ij}^* = \#\{y_{ik}^* = y_j\}. \tag{15}$$

With this definition, $\mathbf{Y}_i^*$ is a vector enumerating the representation of each original data point in the $ith$ bootstrap replicate as follows:

$$\mathbf{Y}_i^* = \{Y_{i1}^*, Y_{i2}^*, ..., Y_{iN}^*\}. \tag{16}$$

Further, the average representation of the original spectral density value $y_j$ across the $B$ bootstrap replicates is given by the following:

$$\bar{Y}_j^* = \frac{1}{B} \sum_{i=1}^{B} Y_{ij}^*. \tag{17}$$

The covariance between the representation of the $jth$ spectral density value and the $kth$ model-free parameter value across $B$ bootstrap replicates is given by the following:

$$\hat{cov}_{jk} = \frac{1}{B} \sum_{i=1}^{B} \left( Y_{ij}^* - \bar{Y}_j^* \right) \left( \tilde{\mu}_{ik}^* - \tilde{\mu}_k \right). \tag{18}$$

Finally, the smoothed estimate of the standard deviation for the $kth$ model-free parameter is calculated from the following expression:

$$\tilde{\sigma}_k = \left[ \frac{1}{N} \sum_{j=1}^{N} c\hat{o}v_{jk}^2 \right]^{1/2} . \tag{19}$$

In bootstrap aggregation, the reported results consist of the smoothed estimators $\{\tilde{\mu}_k\}$ and $\{\tilde{\sigma}_k\}$ incorporating the effects of model-selection uncertainty. As noted by Efron, $\tilde{\sigma}_k \leq \hat{\sigma}_k^u$, in which $\hat{\sigma}_k^u$ is obtained using Eq. 8 naively applied to the bootstrap aggregated data (rather than to data analyzed with a fixed model as above) (Efron, 2014).

## 3  Methods

Backbone amide $^{15}$N spin relaxation data have been reported at $G = 4$ $^1$H static magnetic fields of 600, 700, 800, and 900 MHz for the bZip domain of the *S. cerevisiae* transcription factor GCN4 by Gill and coworkers (Gill et al., 2016). Experimental values of $R_1$, $R_2$, and the steady-state $NOE$ measured at each magnetic field for each residue were converted to spectral density values using the following expressions (Farrow et al., 1995; Gill et al., 2016):

$$J(0.87\omega_H) = \frac{4}{5d_{NH}^2}\sigma_{NH} \tag{20}$$

$$J(\omega_N) = \frac{4(R_1 - 1.249\sigma_{NH})}{3d_{NH}^2 + 4c_{NH}^2} \tag{21}$$

$$J(0) = \frac{6(R_2 - 0.5R_1 - 0.454\sigma_{NH})}{3d_{NH}^2 + 4c_{NH}^2}, \tag{22}$$

in which $\sigma_{NH} = (NOE-1)R_1\gamma_N\gamma_H^{-1}$, $d_{NH} = (\mu_0/4\pi)\hbar\gamma_H\gamma_N r_{NH}^{-3}$, $c_{NH} = 3^{-1/2}\Delta\sigma\omega_N$, $r_{NH} = 0.102$ nm is the N-H bond length, and $\Delta\sigma = -172$ ppm is the $^{15}$N chemical shift anisotropy. A single value of $J(0)$ was obtained for each residue as the weighted mean (using propagated experimental uncertainties) of the values obtained from the $G$ static magnetic fields. The uncertainty in the mean $J(0)$ was obtained by jackknife simulations. For each residue, the spectral density values used for model fitting consist of the mean $J(0)$, $G$ values of $J(\omega_N)$ and $G$ values of $J(0.87\omega_H)$, for a total of 9 data points.

As noted above, the $^{15}$N spectral density values for each backbone amide consist of $G = 4$ values of each of $J(0)$, $J(\omega_N)$ and $J(0.87\omega_H)$. Random sampling with replacement from the $N = 12$ values to generate bootstrap samples, as normally applied, could result in samples in which the relative numbers of spectral density values from each class are highly skewed. For example, a bootstrap sample could be generated without any $J(0)$ values, leading to very anomalous fitted parameters. At the other extreme, random sampling with replacement could result in samples in which a single value was highly over-represented. For example, a bootstrap sample could be generated in which one particular $J(0)$ value is represented exclusively.

To avoid such highly unrepresentive possibilities, bootstrap samples were generated by enumerating the $19^3 = 6859$ possible arrangements in which at most two spectral density values from each set of $J(0)$, $J(\omega_N)$ and $J(0.87\omega_H)$ are duplicated. The 19 possible arrangements of the $G = 4$ indices $\{1, 2, 3, 4\}$ and corresponding $Y_{ij}$ for selecting bootstrap samples of $J(0)$, $J(\omega_N)$ and $J(0.87\omega_H)$ are shown in Table 1. In this Table, $p_{ij}$ is a pointer vector selecting data from a particular set of spectral density

**Table 1.** Bootstrap Selections

| i | $p_{ij}$ | $Y_{ij}^*$ | i | $p_{ij}$ | $Y_{ij}^*$ |
|---|---|---|---|---|---|
| 1 | [1,2,3,4] | [1,1,1,1] | 11 | [4,2,3,4] | [0,1,1,2] |
| 2 | [1,1,3,4] | [2,0,1,1] | 12 | [1,4,3,4] | [1,0,1,2] |
| 3 | [1,2,1,4] | [2,1,0,1] | 13 | [1,2,4,4] | [1,1,0,2] |
| 4 | [1,2,3,1] | [2,1,1,0] | 14 | [1,1,2,2] | [2,2,0,0] |
| 5 | [2,2,3,4] | [0,2,1,1] | 15 | [1,1,3,3] | [2,0,2,0] |
| 6 | [1,2,2,4] | [1,2,0,1] | 16 | [1,1,4,4] | [2,0,0,2] |
| 7 | [1,2,3,2] | [1,2,1,0] | 17 | [2,2,3,3] | [0,2,2,0] |
| 8 | [3,2,3,4] | [0,1,2,1] | 18 | [2,2,4,4] | [0,2,0,2] |
| 9 | [1,3,3,4] | [1,0,2,1] | 19 | [3,3,4,4] | [0,0,2,2] |
| 10 | [1,2,3,3] | [1,1,2,0] | | | |

values. For example $p_{4j} = [1, 2, 3, 1]$; applying this pointer to the set of $J(0)$ values would select the $J(0)$ values obtained at 600 ($\times 2$), 700, and 800 MHz. The corresponding counter vector $Y_{4j}^* = [2, 1, 1, 0]$ is the numbers of times $J(0)$ values recorded at the different fields were sampled. The process would be repeated for the other sets of spectral density values. For example, the $1260th$ bootstrap sample uses $p_{4j}$ to select $J(0)$, $p_{10j}$ to select $J(\omega_N)$, and $p_{6j}$ to select $J(0.87\omega_H)$. The full vector $Y_{ij}^*$ of length $N = 12$ is obtained by concatenating the individual $Y_{4j}^*$, $Y_{10j}^*$, and $Y_{6j}^*$ vectors from the table. With this procedure, the first bootstrap sample is identical to the original data. The mean and uncertainty was determined for $J(0)$ for each bootstrap sample as described above for the original data so that fitting of bootstrap samples was performed in the same fashion as for the original data.

The data were analyzed by three procedures. First, a conventional analysis, Eq. (4), was performed in which optimal models $t_1$ and model parameters $\{\hat{\mu}_k\}$ were determined for each amino acid residue (for which data were available) using $AIC_C$. The uncertainties in model parameters, denoted $\{\hat{\sigma}_k\}$, were determined by 500 Monte Carlo simulations using the measured experimental uncertainties in the spectral density values (Gill et al., 2016). Second, the optimal model was determined as in the first procedure, but the uncertainties in model parameters, $\{\hat{\sigma}_k^*\}$, were determined by the conventional bootstrap, using Eq. (8). In both of these approaches, error estimates were obtained while fixing the model for each Monte Carlo or bootstrap sample as the optimal model $t_1$ selected against the original data. Third, the smoothed model parameters $\{\tilde{\mu}_k\}$ and uncertainties $\{\tilde{\sigma}_k\}$ were determined by bootstrap aggregation using Eqs. (10) and (19), respectively. In this approach, the optimal model and parameters were determined individually for each bootstrap sample as in Eq. (9). A flowchart outlining the process of performing bootstrap aggregation is shown in Figure 1. Both models 2 and 3 contain a single generalized order parameter and a single internal effective correlation time. The model selection strategy employed herein assigns model 2 if the internal correlation is $< 0.15$ ns and model 3 if the internal correlation time is $\geq 0.15$ ns (vide infra).

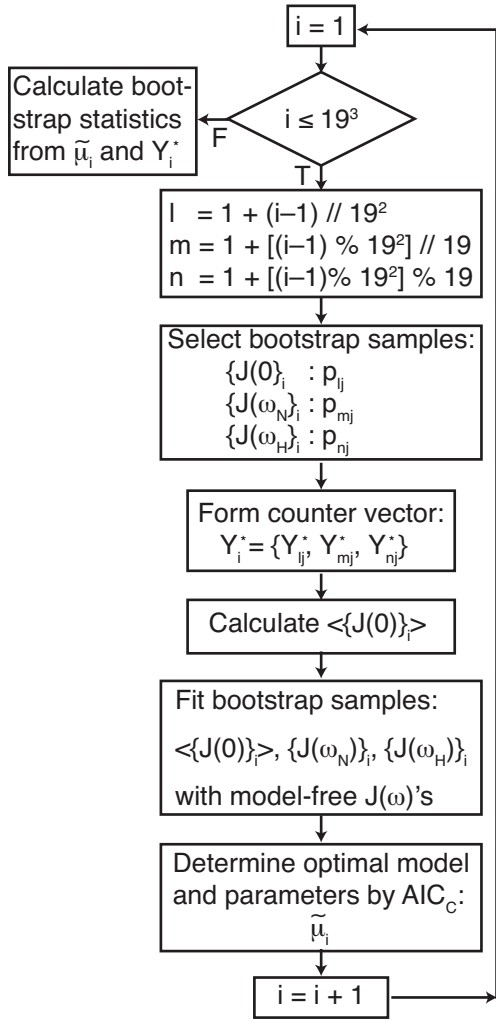

**Figure 1.** Flowchart for bootstrap aggregation for the model-free formalism. Indices $l$, $m$, and $n$ are determined for each bootstrap sample from the index $i$ using modulo arithmatic in which // represents floor division and % is the modulo (remainder) operation. The three indices $l$, $m$, and $n$ select pointer and counter vectors from Table 1. The three pointer vectors are used to generate bootstrap samples for $J(0)$, $J(\omega_N)$, and $J(\omega_H)$. The three counter vectors are concatenated to form $Y_i^*$.

Values of a local $\tau_m$ were optimized for each residue in the well-ordered coil-coil domain of the protein (residues 26-55). Values of $\tau_m$ for residues in the basic region (residues 3–25) and disordered C-terminus (residues 56–58) were fixed at 17.5 ns, the average value obtained for ordered residues. A similar approach was used by Gill and coworkers in the original analysis of the relaxation data (Gill et al., 2016). Local values of $\tau_m$ can be used to determine the overall rotational correlation time or diffusion tensor by established methods (Lee et al., 1997). Alternatively, the fitting process could be modified to globally optimize the overall rotational correlation time or diffusion tensor while independently optimizing generalized order parameters and correlation times for individual residues (Mandel et al., 1995). In this scenario, bootstrap aggregation for the internal dynamical parameters would be performed by the same approach as used herein.

## 4  Results

The results of the conventional analysis using $AIC_C$ for model-selection and Monte Carlo error estimation are shown in Figure 2. Each of the Monte Carlo simulations was analyzed using the optimal model determined from the original data. The optimal parameters differ slightly from those reported by Gill and coworkers, because the present approach used a different spectral density function and model-selection method compared to the earlier work (Gill et al., 2016). The results of the conventional analysis using $AIC_C$ for model-selection and bootstrap resampling for error estimation are shown in Figure 3. Each of the bootstrap data sets was analyzed using the optimal model determined from the original data. The results for bootstrap aggregation using $AIC_C$ to determine the optimal model for each bootstrap sample are shown in Figure 4. The boostrap-aggregated smoothed model-free parameters were calculated using Eq. (10) and the smoothed parameter uncertainties were calculated using Eq. (19).

Bootstrap simulations in which a single optimal model is utilized provide an alternative to Monte Carlo simulations for estimation of (unsmoothed) parameter uncertainties. The uncertainties in $\hat{\sigma}(S^2)$ obtained from Monte Carlo simulations and $\hat{\sigma}^*(S^2)$ obtained from conventional bootstrap simulations are compared in Figure 5a. The uncertainties have approximately the same range, but are uncorrelated with each other. These results suggest the non-parametric bootstrap samples simulate the actual data distribution in comparable manner as the parametric Monte Carlo simulations, but without assuming a normal distribution of spectral density values. The smoothed parameter uncertainty obtained from Eq. (19) is compared to the uncertainties from Monte Carlo simulations in Fig. 5b. The increase in $\tilde{\sigma}(S^2)$ compared to $\hat{\sigma}(S^2)$ reflects the effect of model-selection uncertainty. As noted by Efron, the estimate of smoothed parameter uncertainty obtained from Eq. (19) is smaller than the naive estimate obtained by applying Eq. (6) to the aggregated bootstrap samples (Efron, 2014). To illustrate the advantage of Eq. (19), Fig. 5c compares $\hat{\sigma}^u(S^2)$ obtained from Eq. (8) and $\tilde{\sigma}(S^2)$ obtained from Eq.(19). Similar trends are observed for other model-free parameters (not shown).

The performance of the conventional analysis, in which a single optimal model is chosen, and bootstrap aggregation, in which parameter values are smoothed over all models, are illustrated for particular residues Arg 11, Arg 26, and Asp 32. Table 2 shows the values of $AIC_C$ for each model fit to the original spectral density and the percentage that each model was chosen in the bootstrap aggregation. Table 3 shows the optimized model-free parameters for each model fit to the original spectral

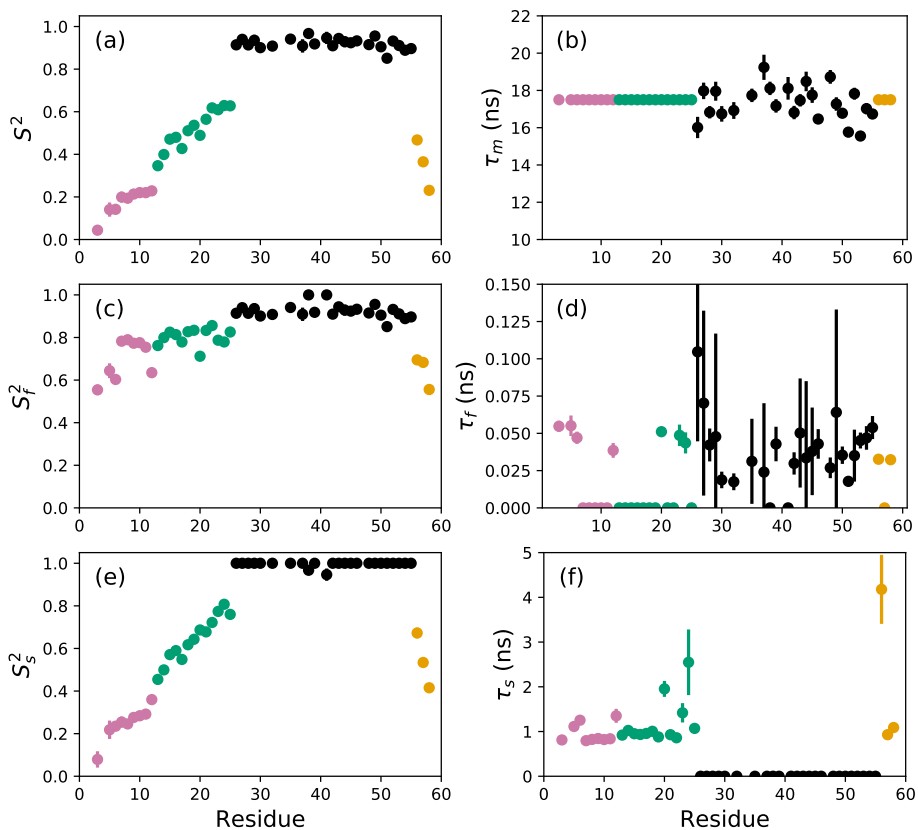

**Figure 2.** Model-free parameters from conventional model selection using $AIC_C$ and 500 Monte Carlo simulations to determine parameter uncertainties. Values of $S^2$, $\tau_m$, $S_f^2$, $\tau_f$, $S_s^2$, and $\tau_s$ are plotted vs. residue number. Overall correlation times ($\tau_m$) were determined individually for residues in the coiled-coil region (black), while $\tau_m$ was fixed at 17.5 ns for residues in the basic region and C-terminus. Regions of the protein are colored as basic region 1 (residues 3–12) (reddish-purple), basic region 2 (residues 13–25) (green), coiled-coil (residues 26–55) (black) , and disordered C-terminus (residues 56–58) (orange) (Gill et al., 2016).

density data and the smoothed model-free parameters obtained by bootstrap aggregation. The optimal single model selected by $AIC_C$ is highlighted with an asterisk.

To further illustrate bootstrap aggregation for Arg 11, Arg 26, and Asp 32, Figures 6, 7, and 8 show the distributions of model-free parameters determined from the optimal model for each bootstrap sample. The calculated spectral density function for bootstrap aggregation is compared to the fitted spectral density functions for each model in Figures 9, 10, and 11.

## 5 Discussion

The difficulties posed by conventional model-selection strategies, in which a single optimal model is chosen using $AIC_C$ or other fitness statistic, are illustrated for the bZip domain of GCN4 in Fig. 2. In particular, some residues in the basic region

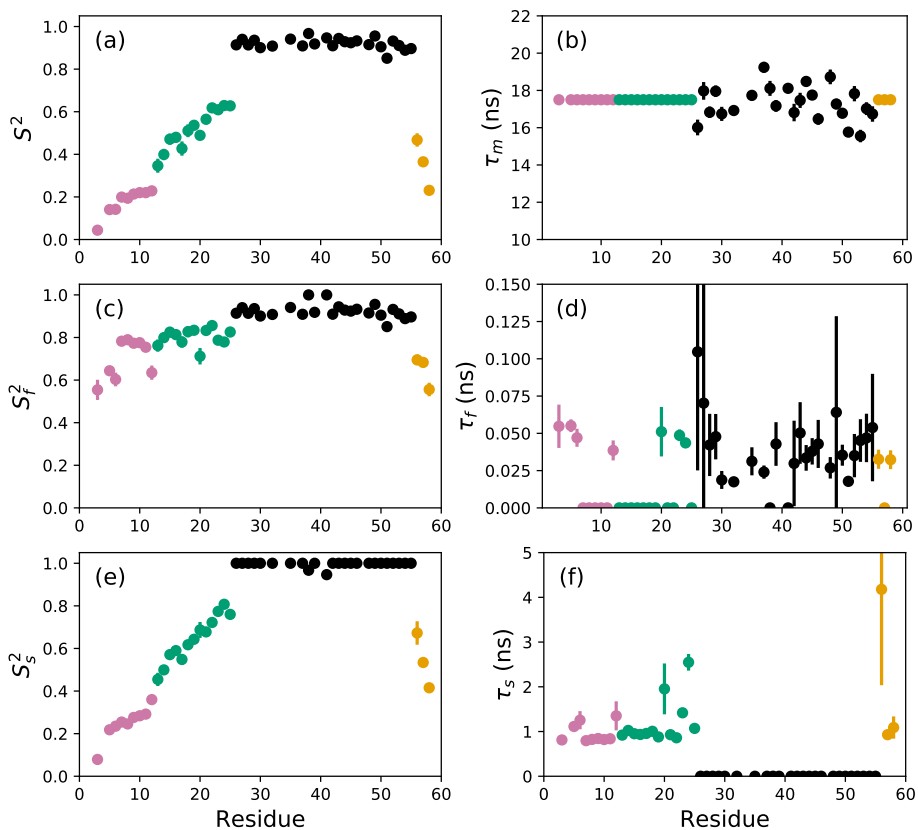

**Figure 3.** Model-free parameters from conventional model selection using $AIC_C$ and bootstrap resampling to determine parameter uncertainties. Values of $S^2$, $\tau_m$, $S_f^2$, $\tau_f$, $S_s^2$, and $\tau_s$ are plotted vs. residue number. Parameter values are identical as in Figure (2), but the uncertainty estimates differ. Regions of the protein are colored as basic region 1 (residues 3–12) (reddish-purple), basic region 2 (residues 13–25) (green), coiled-coil (residues 26–55) (black) , and disordered C-terminus (residues 56–58) (orange) (Gill et al., 2016)

(residues 3-25) are analyzed using model 4, in which $\tau_f = 0$ and other residues are analyzed with model 5, in which $\tau_f > 0$.
The resulting values of the other model-free parameters are systematically affected depending on whether or not $\tau_f = 0$. These systematic effects are evident most clearly in the scatter in $S_f^2$ and $\tau_s$ for residues in the basic region. The advantages of bootstrap aggregation in smoothing over variability in model selection is evident in Fig. 4, in which the residue-to-residue variability of the model-free parameters is reduced. Thus, the distributions of $\tau_f$ and $\tau_s$ are much more uniform within the four regions of the protein, suggesting rather uniform time-scale processes in each sub-domain. The similarity in the distributions for $\hat{\sigma}(S^2)$ and $\hat{\sigma}^*(S^2)$, shown in Fig. 5a, indicates that the bootstrap procedure adequately samples the distribution of parameter values. That is, the reduction in parameter variablility from bootstrap aggregation does not result from restricted sampling.

The results shown for residue Arg 11 in Tables 2 and 3 and Figs. 6 and 9 illustrate the mechanics behind bootstrap aggregation. The original optimization against the measured data yielded $AIC_C$ values of 33.3 for model 4 and 34.1 for model 5. The conventional analysis then selects model 4 (with $\tau_f = 0$) as optimal, even though $AIC_C$ for model 5 is only slightly larger.

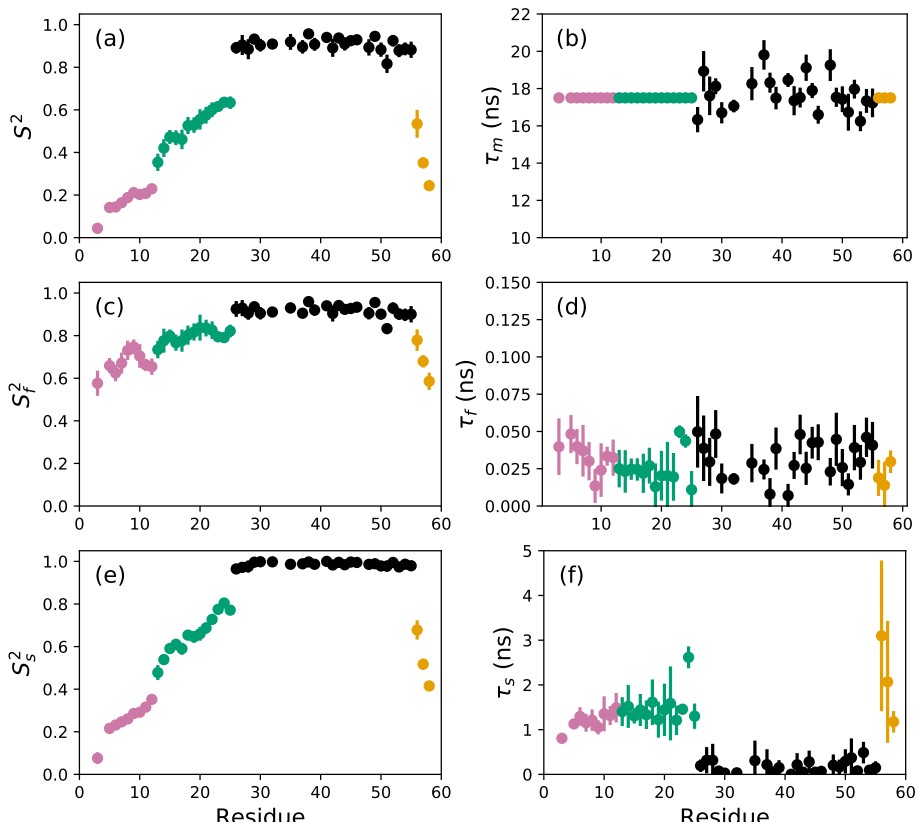

**Figure 4.** Smoothed model-free parameters from bootstrap aggregation to determined smoothed parameter estimates and uncertainties. Values of $S^2$, $\tau_m$, $S_f^2$, $\tau_f$, $S_s^2$, and $\tau_s$ are plotted vs. residue number. Regions of the protein are colored as basic region 1 (residues 3–12) (reddish-purple), basic region 2 (residues 13–25) (green), coiled-coil (residues 26–55) (black) , and disordered C-terminus (residues 56–58) (orange) (Gill et al., 2016)

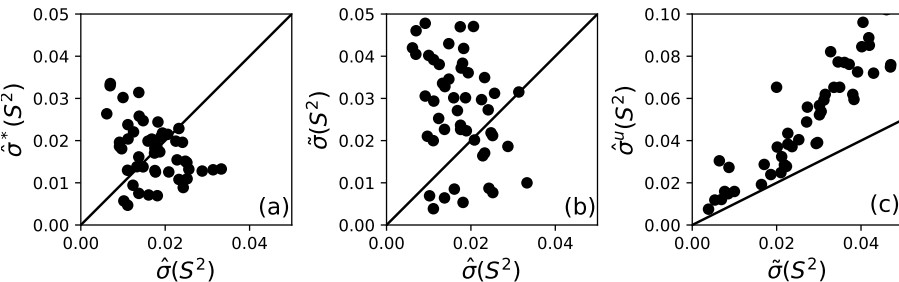

**Figure 5.** Comparison of model-free parameter uncertainties. (a) Uncertainties for $S^2$ calculated from Monte Carlo, $\hat{\sigma}_k$ and bootstrap simulations, $\hat{\sigma}_k^*$, for a single optimal model. (b) Uncertainties for $S^2$ calculated from Monte Carlo simulations for a single optimal model and smoothed $\tilde{\sigma}_k$ calculated from bootstrap aggregation. (c) Uncertainties $\hat{\sigma}_k^u$ and $\tilde{\sigma}$ for $S^2$ calculated from bootstrap aggregation, illustrating the smaller variability obtained using Eq. (19) for calculation of parameter sample deviations.

**Table 2.** Model Selection for Selected Residues

| Residue | Fit | Model 1 | Model 2 | Model 3 | Model 4 | Model 5 |
|---------|-----|---------|---------|---------|---------|---------|
| Arg 11 | $AIC_C$ | 67.9 | NA | 57.2 | 33.3 | 34.2 |
| | Boot | 0.000 | 0.000 | 0.000 | 0.243 | 0.757 |
| Arg 26 | $AIC_C$ | 39.2 | 23.4 | NA | 33.5 | 56.6 |
| | Boot | 0.000 | 0.566 | 0.316 | 0.096 | 0.022 |
| Asp 32 | $AIC_C$ | 18.4 | 10.3 | NA | 22.3 | 46.2 |
| | Boot | 0.000 | 0.970 | 0.000 | 0.019 | 0.011 |

For each residue, the top line lists the $AIC_C$ values determined by fitting the original data to Models 1-5. The second line enumerates the percentage of bootstrap samples for which the indicated model exhibited the lowest $AIC_C$. Both models 2 and 3 contain a single internal effective correlation time. Model 2 is assigned if this correlation time is < 0.15 ns (and model 3 is not assigned, NA). Model 3 is assigned if this correlation time is ≥ 0.15 ns (and model 2 is not assigned, NA).

**Table 3.** Model-free Parameters for Selected Residues

| Residue | Model | $\tau_m$ | $S^2$ | $S_f^2$ | $S_s^2$ | $\tau_f$ | $\tau_s$ |
|---------|-------|----------|-------|---------|---------|----------|----------|
| Arg 11 | 1 | 17.5(fixed) | 0.886 ± 0.015 | 0.886 ± 0.015 | 1 | 0 | 0 |
| | 3 | 17.5(fixed) | 0.480 ± 0.006 | 1 | 0.480 ± 0.006 | 0 | 0.761 ± 0.011 |
| | 4* | 17.5(fixed) | 0.220 ± 0.017 | 0.754 ± 0.015 | 0.292 ± 0.018 | 0 | 0.838 ± 0.014 |
| | 5 | 17.5(fixed) | 0.211 ± 0.017 | 0.646 ± 0.022 | 0.326 ± 0.020 | 0.036 ± 0.004 | 1.13 ± 0.09 |
| | Smooth | 17.5(fixed) | 0.208 ± 0.005 | 0.662 ± 0.029 | 0.316 ± 0.013 | 0.033 ± 0.006 | 1.31 ± 0.21 |
| Arg 26 | 1 | 14.55 ± 0.48 | 0.954 ± 0.031 | 0.954 ± 0.031 | 1 | 0 | 0 |
| | 2* | 16.01 ± 0.55 | 0.914 ± 0.024 | 0.914 ± 0.024 | 1 | 0.105 ± 0.054 | 0 |
| | 4 | 16.00 ± 0.73 | 0.878 ± 0.038 | 0.935 ± 0.037 | 0.939 ± 0.013 | 0 | 0.274 ± 0.165 |
| | 5 | 17.28 ± 2.69 | 0.812 ± 0.103 | 0.871 ± 0.070 | 0.932 ± 0.057 | 0.030 ± 0.020 | 0.93 ± 1.15 |
| | Smooth | 16.33 ± 0.68 | 0.891 ± 0.027 | 0.925 ± 0.037 | 0.972 ± 0.015 | 0.050 ± 0.024 | 0.19 ± 0.14 |
| Asp 32 | 1 | 16.28 ± 0.39 | 0.944 ± 0.022 | 0.944 ± 0.022 | 1 | 0 | 0 |
| | 2* | 16.92 ± 0.46 | 0.908 ± 0.025 | 0.908 ± 0.025 | 1 | 0.017 +/- 0.016 | 0 |
| | 4 | 16.92 ± 1.31 | 0.908 ± 0.053 | 1.000 ± 0.060 | 0.908 ± 0.040 | 0 | 0.02 ± 0.48 |
| | 5 | 19.58 ± 3.45 | 0.756 ± 0.138 | 0.853 ± 0.074 | 0.887 ± 0.091 | 0.010 ± 0.009 | 8.33 ± 3.18 |
| | Smooth | 17.06 ± 0.34 | 0.909 ± 0.017 | 0.911 ± 0.015 | 0.998 ± 0.005 | 0.018 ± 0.004 | 0.035 ± 0.074 |

For each residue, parameter values for Models 1-5 are calculated from the fit of the original data to the relevant spectral density function, with errors determined by Monte Carlo simulation. The model selected by $AIC_C$ is indicated by *. Smooth values are obtained by averaging the best fit parameter values across bootstrap samples as in Eq.10, with errors determined as indicated in Eqs. 15-19.

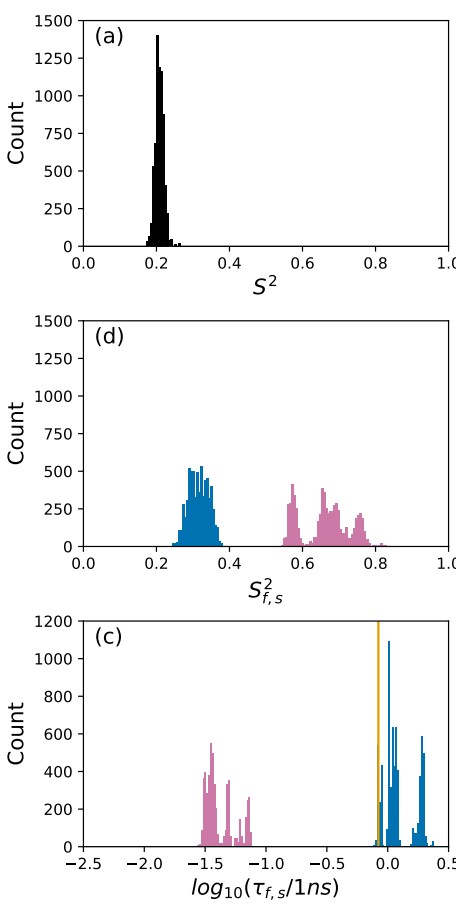

**Figure 6.** Distribution of model-free parameters from bootstrap aggregation for residue Arg 11. Color coding is $S_f^2$ or $\tau_f$ (reddish-purple) and $S_s^2$ or $\tau_s$ (blue). The orange line in (c) indicates the value of $\tau_s$ obtained for the optimal single (unsmoothed) model 4. For clarity, null values of 1 for generalized order parameters and 0 for internal effective correlation times are not shown in the graphs; $\tau_f = 0$ is observed 1664 times.

In contrast the bootstrap analysis suggests that model 4 would be optimal for 24% and model 5 would be optimal for 76% of randomly chosen data, under the assumption that the bootstrap samples represent the underlying distribution of spectral density values. Bootstrap smoothing then averages each model-free parameter over the empirical distributions shown in Fig. 6, with resulting optimized spectral density curves compared to the original experimental data in Fig. 9. The results for model 4 in Table 3 and the corresponding vertical orange line in Fig. 9 shows that the selection of model 4 in the conventional analysis

results in an estimate for $\tau_s$ that is skewed toward the lower boundary of the $\tau_s$ bootstrap distribution.

     The results shown for residue Arg 26 in Tables 2 and 3 and Figs. 7 and 10 illustrate another advantage of bootstrap aggregation. In this case, the original optimization against the measured data yielded an $AIC_C$ value 23.4 of for model 2, substantially smaller than for any other model, implying a single model might be an adequate description for this residue. However, the

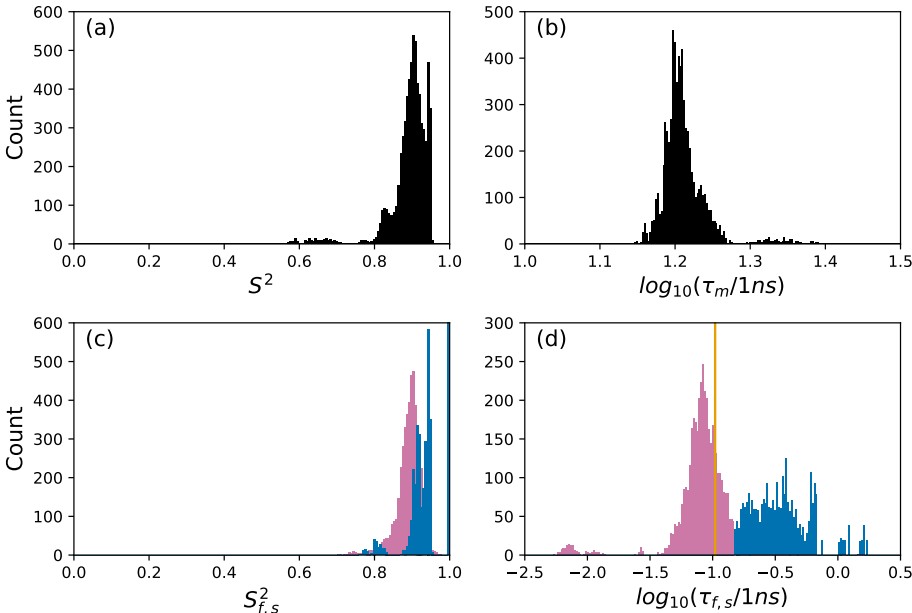

**Figure 7.** Distribution of model-free parameters from bootstrap aggregation for residue Arg 26. Color coding is $S_f^2$ or $\tau_f$ (reddish-purple) and $S_s^2$ or $\tau_s$ (blue). The orange line in (c) indicates the value of $\tau_f$ obtained for the optimal single (unsmoothed) model 2. For clarity, null values of 1 for generalized order parameters and 0 for internal effective correlation times are not shown in the graphs; $S_f^2 = 1$ is observed 2167 times, $S_s^2 = 1$ is observed 3884 times, $\tau_f = 0$ is observed 2823 times, and $\tau_s = 0$ is observed 3884 times.

bootstrap distribution for the internal correlation times is bimodal. The conventional choice of model 2 reults in an estimate
of $\tau_f$ roughly centered in the distrubution, but the smoothed bootstrap estimates identify the presence of two separable time scales for internal motions, one with a mean $0.052 \pm 0.019$ and the other with mean $0.13 \pm 0.08$. Residue 26 is at the juncture between the basic region and coiled-coil motif of the GCN4 bZip domain; consequently, the latter effective internal correlation time might represent a vestige of the more pronounced motions evident in the basic region. The critical value of 0.15 ns used to separate fast from slow motions in the present work was chosen empirically to distinguish the two distributions observed for
residue 26 (and used for all other residues). More sophisticated clustering algorithms could be used to make this distinction between models 2 and 3.

    The results shown for residue Asp 32 in Tables 2 and 3 and Figs. 8 and 11 illustrate a case of strong agreement between the conventional analysis and bootstrap aggregation when a single motional model is strongly favored by the experimental data. The distributions shown in Fig. 8 then represent the variability in model-free parameters across the bootstrap samples. These
results would be comparable to results obtained in Fig. 3, in which the bootstrap samples were used to estimate model-free parameter uncertainties $\hat{\sigma}_k^*$ for a single fixed optimal model.

    The present application of bootstrap aggregation used spin relaxation data recorded at four static magnetic fields. A total of 6859 bootstrap samples were used to calculate smoothed parameter estimates. Data recorded at three static magnetic fields

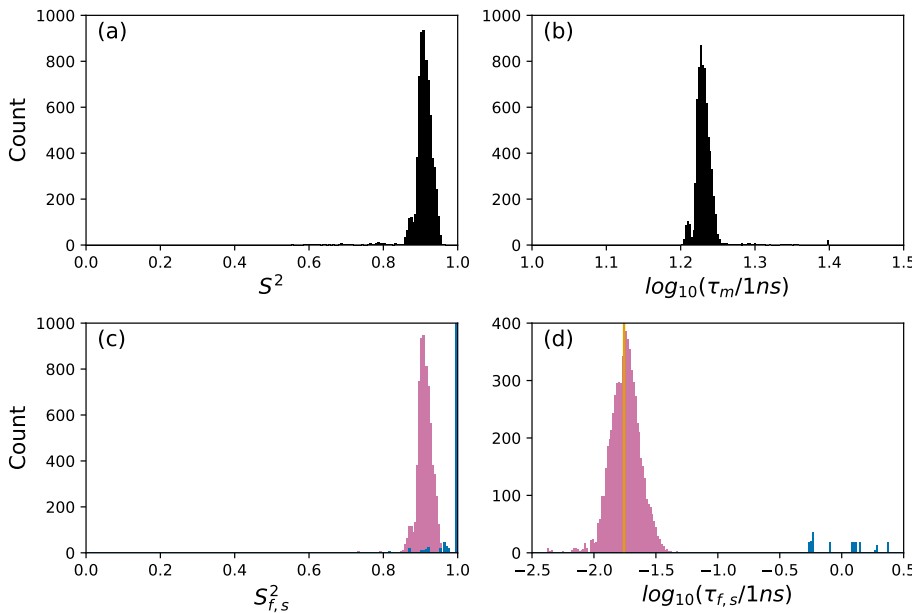

**Figure 8.** Distribution of model-free parameters from bootstrap aggregation for residue Asp 32. Color coding is $S_f^2$ or $\tau_f$ (reddish-purple) and $S_s^2$ or $\tau_s$ (blue). The orange line in (c) indicates the value of $\tau_f$ obtained for the optimal single (unsmoothed) model 2. For clarity, null values of 1 for generalized order parameters and 0 for internal effective correlation times are not shown in the graphs; $S_s^2 = 1$ was observed 6650 times.

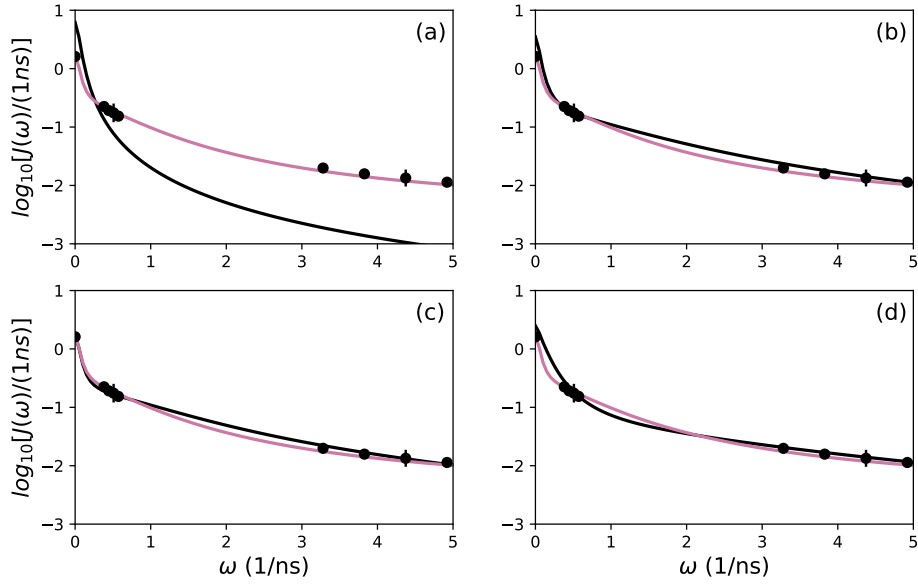

**Figure 9.** Comparison of individual fits for Arg 11 of (a) model 1, (b) model 3, (c) model 4, and (d) model 5 (black lines) or the bootstrap aggregation smoothed fit (reddish-purple line) to (circles) experimental spectral density values.

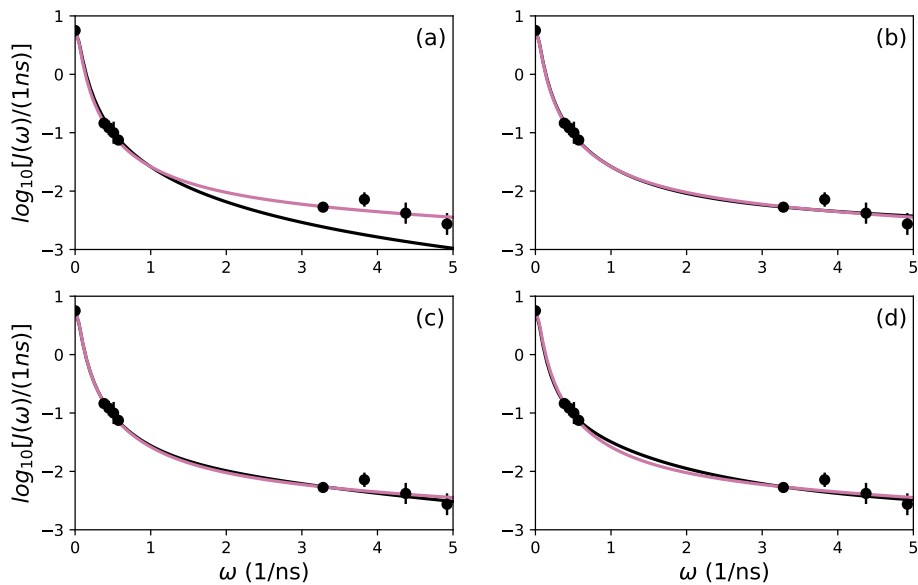

**Figure 10.** Comparison of individual fits for Arg 26 of (a) model 1, (b) model 3, (c) model 4, and (d) model 5 (black lines) or the bootstrap aggregation smoothed fit (reddish-purple line) to (circles) experimental spectral density values.

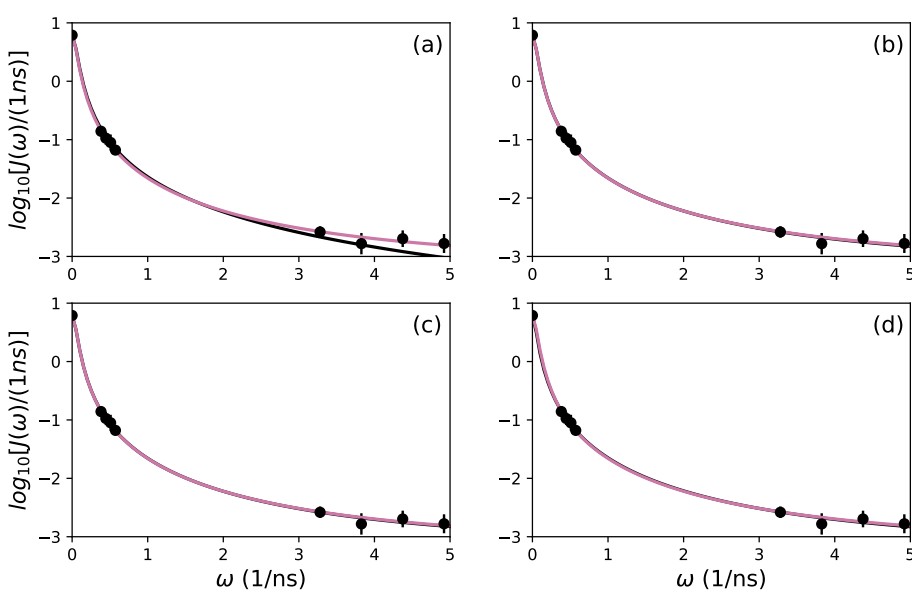

**Figure 11.** Comparison of individual fits for Asp 32 of (a) model 1, (b) model 3, (c) model 4, and (d) model 5 (black lines) or the bootstrap aggregation smoothed fit (reddish-purple line) to (circles) experimental spectral density values.

provides 9 spectral density values, but allows only $7^3 = 343$ bootstrap samples. To test the effect of such a drastic reduction in the size of the bootstrap sample, the relaxation rate constants recorded at 600, 800, and 900 MHz were analyzed for the disordered basic region (residues 3-25). This preserves the same range of sampled frequencies as for the original analysis, but only 7 spectral density values are obtained for each residue, after averaging the three values of $J(0)$. The smaller number of spectral density values results in smaller numbers of degrees of freedom when fitting the model-free spectral density models. As a consequence, only models 3 and 4 were selected for the basic region in the conventional analysis; essentially, the data were not sufficient to determine $\tau_f$ and $\tau_s$ simultaneously (model 5). Nonetheless, bootstrap aggregation was effective in smoothing the effects of model selection error between models 3 and 4, even with only 343 bootstrap samples (not shown). A number of studies have investigated the number of model parameters that can be determined from backbone amide $^{15}$N relaxation data recorded at high static magnetic fields (Khan et al., 2015; Gill et al., 2016; Abyzov et al., 2016) The present results suggest that measurements at four static magnetic fields are required to fully statistically characterize the information content of such measurements within the extended model-free formalism.

## 6 Conclusions

Model-selection error is a classical problem in statistics and has been recognized as a concern in the model-free analysis of NMR spin relaxation data since the work of d'Auvergne and Gooley (d'Auvergne and Gooley, 2007, 2008a, b). Bootstrap aggregation has emerged as a powerful approach for incorporating selection error into statistical model-building (Buja and Stuetzle, 2006; Efron, 2014). However, bootstrap aggregation requires sufficient numbers of data points to allow faithful re-sampling of the distribution of the data. This issue is made more serious by the nature of nuclear spin relaxation data: spectral density values for $J(0)$, $J(\omega_N)$ and $J(0.87\omega_H)$ are very different and should not be interchanged by resampling. As shown in the present work, resampling within blocks of spectral density values clustered as $J(0)$, $J(\omega_N)$ and $J(0.87\omega_H)$ recorded at three or four static magnetic fields is sufficient to enable bootstrap aggregation. However, the larger data set available from four static magnetic fields allows more reliable resolution of two internal correlation times, $\tau_f < 0.15$ ns and $\tau_s \geq 0.15$ ns.

Aggregation improves parameter stability by averaging over all models represented in the bootstrap sample. As applied to $^{15}$N spin relaxation data for the bZip domain of GCN4, bootstrap aggregation reduces residue-to-residue variations in optimal model-free parameters, particularly in the partially disordered basic region. Consequently, trends in the conformational dynamics along the polypeptide backbone that reflect actual physical properties of the protein become more evident. Notably, local maxima in generalized order parameters within the basic region (residues 3-25), most evident for residues 8 and 9 and for residues 14 and 15 in Fig. 4, reflect transient populations of helical conformations observed in molecular dynamics simulations (Robustelli et al., 2013). NMR spin relaxation spectroscopy is a powerful approach for interrogating conformational dynamics of biological macromolecules. Bootstrap aggregation, coupled with experimental NMR spin relaxation measurements at multiple static magnetic fields, promises to advance efforts to understand the interplay between conformation and function in biology.

*Code and data availability.* A Jupyter notebook (Python 3.6) is provided for performing all data analyses reported in the publication. The NMR data analyzed in the publication are available at Mendeley Data (http://dx.doi.org/10.17632/vpwz6mrynr.1).

*Author contributions.* A.G.P. conceived the project. Calculations and writing of the paper were performed by T.C. and A.G.P.

*Competing interests.* The authors declare no competing interests.

*Acknowledgements.* This work was supported by National Institutes of Health grant R35 GM130398 (A. G. P.). Some of the work presented here was conducted at the Center on Macromolecular Dynamics by NMR Spectroscopy located at the New York Structural Biology Center, supported by a grant from the NIH National Institute of General Medical Sciences (P41 GM118302). A.G.P. is a member of the New York Structural Biology Center. This paper is dedicated to Prof. Geoffrey Bodenhausen on the occasion of his 70th birthday.

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
