# Peer review of "Bootstrap Aggregation for Model Selection in the Model-free Formalism"

_Magnetic Resonance, 2021_

## Author Comment (AC1)

Referee 1.

We thank the referee for the careful reading of the manuscript.

1.      Could you please comment on the number of field strengths required to successfully implement bootstrap aggregation. Assuming I get the math right, using 3 field strengths, the total number of possible samples is $7^3 = 343$, which seems to be at the low end of the number of samples normally employed, but perhaps this would suffice?

   **Response**: The referee's math is correct. Reducing the number of field strengths from four to three has three potential consequences for data analysis:

   a.  Depending on the three fields chosen, the total range of frequencies can be affected. For example, three fields in the range 600 MHz – 1.2 GHz are likely to be more powerful than three fields in the range 600 MHz to 800 MHz.
   b.  Reducing the number of fields reduces the number of data points and thus the number of statistical degrees of freedom. The number of degrees of freedom influences data analysis, whether conventional approaches or the bootstrap aggregation method are used.
   c.  Most pertinent to the referee's point, three fields provide only 343 bootstrap samples, compared to the 6859 samples obtained for four fields – a 20-fold increase provided by one additional field.

   To address the referee's question, the analysis for the residues in the basic region (the disordered sites) was repeated using only the 600 MHz, 800 MHz and 900 MHz data, dropping the 700 MHz data, but keeping the same total range of frequencies. The reduced size of the data set resulted in simpler models being selected for individual residues in the basic (disordered region), because of the reduced number of spectral density values available for fitting. Essentially, the data were not sufficient to determine both $\tau_f$ and $\tau_s$, so most residues were fit with only $\tau_s$ (models 3 and 4). However, even with the reduced number of bootstrap samples, bootstrap aggregation was effective in smoothing the effects of model selection error arising from choices between these two models.

2.      How do you determine the correlation time for overall rotational diffusion, tau_m (in the case of a globular protein)? The suggested protocol fits tau_m individually for each residue. Do you forego the concept of fitting a global tau_m as part of the MF fits? And subsequently fit a rotational diffusion model (isotropic or anisotropic) to these individual values (while taking into account the orientation of the HN bond vectors in the molecular frame in the case of anistropic models)?

**Response**: As noted by the referee, the approach adopted in the manuscript fits individual values of $\tau_m$ for residues in the ordered domain of the protein. This approach was adopted to correspond to the strategy used in the earlier paper by Gill and coworkers in which the relaxation rates were originally reported (Gill, et al., Phys Chem Chem Phys, 18, 5839–49, 2016). In a subsequent step, the global diffusion tensor could be determined from the values of $\tau_m$ as described in many

publications (for example, Lee et al., J. Biomol. NMR, 9, 287–298, 1997). This approach has the advantage of decoupling the determination of internal motional parameters from overall rotational diffusion (as discussed by d'Auvergene and Gooley (Mol. Biosyst., 3, 483–94, 2007), but increases the number of fitted parameters (in the simplest case, multiple local values of $\tau_m$, one for each residue, must be fit rather than one global value). The bootstrap aggregation approach can be applied equally well in a data analysis strategy that optimizes a global rotational diffusion time or tensor. The two approaches for determining the overall rotational diffusion model differ how 'structural noise' arising from the reference N-H bond vector orientations (for example, in an x-ray crystal structure) differ from the time/ensemble average in solution. The present manuscript is not intended to assess the relative merits of either approach.

Minor points:

line 19, Suggestion: spell out Akaike and Bayesian Information Criterion when introducing AIC and BIC.

> **Response**: Done

l. 145-148, the mean J(0) is presumably only used in the conventional MF protocol with MC error analysis(?) This should be stated here to avoid confusion.

> **Response**: The mean J(0) was used in all analyses. However, the mean was the mean of the bootstrap samples for J(0) in the bootstrap analyses. We have clarified this point in the text.

The bootstrap aggregation protocol is well described on p. 6, but I still feel that it might be beneficial to include flow-chart type figure outlining the construction of the bootstrap sample datasets.

> **Response**: We have prepared a flowchart for the revised manuscript.

The tables are not easily interpreted without referring back to the text. Please add footnotes to define p_ij and Y_ij in words (Table 1). Please add text to indicate that "Smooth" refers to the percentage of selected models in Tables 2-4.

> **Response**: We have clarified the tables as requested by all referees.

Typos: "paramaters" (l. 86); "interogating conformional" (l. 250)

> **Response**: Thank you for the careful reading.

---

## Author Comment (AC2)

Referee 2.

We thank the referee for the careful reading of the manuscript.

Regarding the more general practical utility of the proposed sampling protocol, while experimental relaxation data collected at four magnetic field strengths yields 6859 suitably filtered combinations of bootstrap samples, as noted by the first reviewer, the robustness of the statistical analysis may appreciably decline when this value drops to 343 for three magnetic field strengths, and presumably will decrease significantly further when it drops to only 27 for data from two magnetic field strengths.

> **Response**: The reduction to three fields has been discussed in the response to referee 1. Further reduction to two fields would not allow any reasonable resampling.

A key step in the proposed joint refinement process calculates each of the averaged dynamical parameters by summing over the estimates obtained from each of the five spectral density representations being utilized, as weighted by how often each of these five models have been selected (Eq. 10). A potential concern over this approach arises from the fact that while the same set of symbols ($\tau_m$, $S_f^2$, $\tau_f$, $S_s^2$, $\tau_s$) are utilized in each of the five dynamical models used, the functional significance of each symbol is defined within the context of the specific equation being used.

> **Response**: The referee is correct in the sense that if only one internal time scale is fit, for example, $\tau_s$, then the optimized value of the parameter will average over both fast and slow time scales (in some complex manner), whereas if both $\tau_f$ and $\tau_s$ are included in the model, then some partitioning of the time scales occurs. This issue is exactly what the bootstrap aggregation procedure addresses. That said, care must be taken in the interpretation of the fitted parameters for different models when aggregating results. In the present application, models that incorporated a single internal correlation time were partitioned between $\tau_f$ (model 2) and $\tau_s$ (model 3) based on an empirical criterion, as described in the paper. One can imagine situations in which deciding how to perform the averaging between model parameters would be a more difficult question.

Each of these five model equations that are used to represent the spectral density function is capable of accurately fitting only a small subset of the physically plausible spectral density curves. Systematic bias can potentially arise not only with respect to a given dynamics parameter being utilized in distinct model representations but also as a result of the inadequacy with which each of the five model spectral density equations are capable of representing the physical dynamics of the system. While such biasing effects are surely diminished for Model 4 and 5 which incorporate four and five adjustable parameters, respectively, more promising might be the utilization of alternative model equations for the spectral density function that can more robustly represent the range of motion occurring in protein molecules which utilize a smaller set of adjustable parameters for optimization against experimental relaxation data.

> **Response.** The referee raises important question concerning the nature of the models used to fit relaxation (or any) data. The merits and limitations of the "model-free" approach to

analyzing spin relaxation data have been discussed beginning with Lipari and Szabo in their original papers and many others subsequently. A number of alternative models for the spectral density function have been proposed, including a number of interesting distribution functions for correlation times, and molecular dynamics simulations are becoming more capable of directly estimating relaxation rate constants. The introductory paragraph of the paper introduced some of these alternative approaches, but was not comprehensive. This paragraph has been expanded in the revision to include some additional recent work in this arena. Nonetheless, the model-free strategy remains the most widespread approach used for analysis of spin relaxation data. The present paper addresses a weakness in this approach: model selection error, and hence will be useful to a large number of researchers. At the same time, bootstrap aggregation is a general approach for treating model selection error and can be applied in the context of other approaches for analyzing relaxation data.

---

## Author Comment (AC3)

Referee 3.

We thank the referee for the careful reading of the manuscript.

Like the other reviewers, discussion needs to address the issues of using two or three fields. Data at two fields are the most commonly acquired and so would this method be inapproriate or do the authors have alternate approaches/ideas.

**Response**: The reduction to three fields has been discussed in the response to referee 1. Further reduction to two fields would not allow any reasonable resampling.

The paper is well-written and accessible to most in the field. A good balance of theory, method and application. I agree that the tables need better notes - for example the description of the colour scheme in Figure 1 has to be repeated in Fig 2 and 3; better descriptive footnotes in Table 2,3,4 (and actually I think this could be put into a single table with residue first column and clear breaks); possibly the same for Tables 5,6,7.

**Response**: We have combined the tables and added information to both the tables and figures as requested.

Very few errors found. Including the detected typos line 199 "highlighted"

**Response**: The typographical error has been corrected.